# Perceived Coercion of Persons with Mental Illness Living in a Community

**DOI:** 10.3390/ijerph18052290

**Published:** 2021-02-25

**Authors:** Min Hwa Lee, Mi Kyung Seo

**Affiliations:** 1Department of Social Welfare, Mokpo National University, Muan 58554, Korea; hwasuri79@gmail.com; 2Department of Social Welfare, Gyeongsang National University, Jinju 52828, Korea

**Keywords:** perceived coercion, therapeutic relationship, therapeutic satisfaction, life satisfaction

## Abstract

*Aims*: The purpose of this study is to analyze the effect of the perceived coercion of people with mental illness living in a community on their therapeutic satisfaction and life satisfaction, mediated by therapeutic relationships. *Methods:* We evaluated several clinical variables (symptoms, psychosocial functioning, and insight), levels of perceived coercion, therapeutic relationships, therapeutic satisfaction, and life satisfaction in 185 people with mental illness (Mean age = 47.99, standard deviation (SD) = 12.72, male 53.0%, female 45.9%) who live in the community and use community-based mental health programs. The data collected were analyzed to test the proposed hypotheses using structural equation modeling (SEM). *Results:* The correlation analysis of all variables showed that clinical variables had statistically significant correlations with therapeutic relationship, therapeutic satisfaction, and life satisfaction, but no significant correlation with perceived coercion. Furthermore, perceived coercion was found to have significant predictive power for treatment satisfaction and life satisfaction mediated by therapeutic relationship. Specifically, the lower the perceived coercion, the better the therapeutic relationship. This, in turn, has a positive effect on the therapeutic satisfaction and life satisfaction of participants. *Conclusions:* Based on these findings, we suggest strategies to minimize coercion in a community.

## 1. Introduction

The use of coercive treatment in the mental health field has led to an ethical dilemma for professionals attempting to balance therapeutic gains with respect for individuals. The main conflict arises between the position that coercive intervention is necessary for the optimal treatment of patients, and the position that treatment against an individual’s will, no matter how effective, cannot be expected to provide positive outcomes [1]. The purposes of modern mental health practices are to promote recovery and emphasize respect for the rights and life satisfaction of persons with mental illness. Even if there are some variations, mental healthcare law in most countries permits involuntary hospitalization, deprivation of liberty, detention, interference with privacy, and restrictions on freedom of movement [2].

Coercion has been defined as a situation in which an individual lacks autonomy, or as an event occurring when one party (an agent) exercises control over another (a target) by constraining the target’s freedom or self-determination [3]. Coercion includes not only force but also acts of manipulation and persuasion that do not involve force [4,5]. The three types of coercion generally experienced by psychiatric patients are legal status, coercive measures, and perceived coercion [6]. Unlike the former two types of coercion, which imply objective forms of coercion, perceived coercion refers to what the individual subjectively perceives. Perceived coercion is considered a particularly important factor for predicting results in cognitive aspects, such as treatment adherence and insight, since it emphasizes the subjective framework of the individual [3,7,8]. In the field of mental health, coercion tends to be justified for the safety, appropriate treatment, and long-term recovery of patients who are incompetent, dangerous, or not aware of their need for treatment due to a lack of insight. The logic that justifies these situations is paternalism. Paternalism is defined as, “the intentional overriding of a person’s known preference or actions by another person, where the person who overrides justifies the action by the goal of benefiting or avoiding harm to the person whose will was overridden” [9]. However, there is ongoing debate as to whether the principles of paternalism justify the use of coercion. Some studies support the paternalistic assumptions that coercion in hospitalization has reduced the possibility of readmission [10] and enhanced psychosocial functioning [11]. Most of these studies have focused on legal status and involuntary hospitalization. Conversely, other studies have predicted that coercion leads to negative outcomes. These studies have mainly focused on perceived coercion. The coercion perceived by the patient during hospitalization did not contribute to raising the awareness of the need for treatment and adherence to the treatment [1,12,13]. Instead, it negatively affected psychosocial functioning, psychiatric symptoms, insight, and therapeutic relationships [7,14,15]. Therefore, perceived coercion, which is the subjective coercive experience of the parties, mostly refutes paternalism.

Due to deinstitutionalization, treatment facilities are shifting from within hospitals to within the community, and case management and assertive community treatment are being increasingly emphasized for the recovery of people with mental illness. While these community-based programs emphasize the therapeutic involvement and choice of patients, for those who do not want to participate in the program, coercion is enacted by restricting freedom and legally exerting control. In hospitals, coercion was justified to prevent harm to both the individuals themselves and to others. However, in the community the coercion is justified for considerations of welfare benefits, therapeutic benefits, and policy effectiveness through involuntary outpatient treatment orders, assertive outreach, assertive community treatment, and service based on housing programs [16]. Community treatment orders represent a formal type of coercion in a community. These orders are referred to by different names, including assisted outpatient treatment order, outpatient commitment, and community treatment order (CTO) [17]; for simplicity and consistency, the term CTO will be used in this study. The principal mechanism of a CTO is the use of coercion to encourage adherence to a given mental health intervention =to treat people in the least restrictive environment [18].

Individuals receiving treatment in community settings may experience less physical coercion than those in inpatient settings; however, these patients may perceive psychological coercion from the pressure that is exerted on them by their case managers or family members to comply with treatment [19]. Even if patients are voluntarily receiving local community care rather than CTOs, ongoing contact with the therapist can be perceived as a monitoring power. This intervention may only be superficially collaborative, as clients experience coercion in the form of unwelcomed persistence and ambiguous thresholds for the use of force [20].

Most studies on the coercion experienced by persons with mental illness in the community have focused on CTOs. Studies analyzing the effectiveness of CTOs have reported that CTOs contribute to reduced relapse, reduced hospitalization, and shorter hospital stays [17,21,22]. However, these results may have reflected regression to the mean for clients who were extreme recipients of service [20]. Studies on the perceived coercion of persons with mental illness who are receiving general community treatment rather than compulsory treatment are extremely limited. The perceived coercion of patients in community treatment processes is related to the therapeutic relationship. In other words, the coercion causes patients to feel disrespected and experience pressure in the therapeutic relationship [23]. The therapeutic relationship in the community should be cooperative and mutually reliable. A positive therapeutic relationship has a significant effect on the treatment outcome [24] and is the most powerful predictor of recovery [25,26]. Therefore, it is particularly important to consider the effect of coercion perceived by patients on the therapeutic relationship for recovery.

In Korea, community-based services have expanded substantially since the passing of the Mental Health Act in 1995 and outpatient treatment orders were legislated in 2008. However, outpatient treatment orders are rarely utilized, and mandated hospitalization still accounts for 46.1% of all hospitalizations. This is quite high compared to the corresponding rates of 17% in Germany, 13.5% in the UK, and 12% in Italy [27]. In other words, outpatient treatment orders have not replaced involuntary hospitalization. One of the reasons Korea has such a high involuntary hospitalization rate is its collectivist culture, which leads people to accept authoritarian or paternalistic approaches more easily than those in the West.

Therefore, this study analyzed the effect of the perceived coercion of persons with mental illness who use community mental health services in Korea—on their therapeutic satisfaction and their life satisfaction mediated by the therapeutic relationship. We considered therapeutic satisfaction and life satisfaction because these variables are the main factors that determine recovery from the point of view of a person who has experienced coercion rather than clinical characteristics such as readmission rate, BPRS (Brief Psychiatric Rating Scale), and treatment adherence.

## 2. Method

### 2.1. Participants

Institutional Review Board of Gyeongsang National University approval was received for the study (GIRB-A20-Y-0011). We recruited a total of 185 participants with mental disorders over the age of 20 years. All participants obtained written informed consent to take part in the study. The inclusion criteria for the participants with mental disorders included a diagnosis of schizophrenia or a mood disorder according to DSM-5, living within the community, and maintaining ongoing contact with the community service unit.

Information on the 185 participants is shown in Table 1. Among the participants, 98 (53%) were male and 85 (45.9%) were female, while 2 (1.1%) did not respond to the question about gender. The average age was 47.99 (±12.72); 18 participants were from 20 to 29 (9.7%), 31 were from 30 to 39 (16.8%), 46 were from 40 to 49 (24.9%), 56 were from 50 to 59 (30.3%), and 34 participants were over 60 (18.4%).

The average years of education was 12.07 (±2.86), and 47 (25.4%) of the participants were employed, meaning most of the participants (74.1%) were unemployed. Participants with schizophrenia accounted for 119 (64.3%) of all participants, followed by 44 with major depression (23.8%) and 13 with bipolar disorder (7.0%). Most of the participants lived either with their families (109, 58.9%) or alone (62, 33.5%).

### 2.2. Measures

#### 2.2.1. Perceived Coercion

To measure perceived coercion, this study used the Perceived Coercion Scale of the MacArthur Admission Experience Survey, which has been altered to fit the Korean context by Seo et al. [5]. The scale consists of five items: “I felt free to do what I wanted about (coming to the hospital, taking medication, getting treatment)”; “I chose to (come into the hospital, take medication, get treatment)”; “It was my idea to (come to the hospital, take medication, get treatment)”; “I had a lot of control over whether I (went into the hospital, took medication, got treatment)”; and “I had more influence than anyone else on whether I (came into the hospital, took medication, got treatment)”. Each item was rated on a scale from 1 (completely not true) to 5 (completely true), and all items were measured using reverse scoring, wherein higher scores indicate higher levels of perceived coercion. Cronbach’s α for this scale was 0.745.

#### 2.2.2. Therapeutic Relationship

The therapeutic relationship is defined as the relationship between a clinician and a patient in the process of working toward the goals of recovery with mutual trust and closeness [28]. STAR-P (Scale To Assess the Therapeutic Relationship, Patient version), which was originally developed by McGuire-Snieckus et al. [29] and has since been adapted for use in the Korean context, was used as a therapeutic relationship measure. It consists of 12 items examining the extent to which the patient feels understood by their clinician and how much the patient’s treatment reflects mutually agreeable goals. Nine items were about the general therapeutic relationship, while three items about the difficulty of the relationship were measured using reverse scoring. Each item was rated on a scale from 1 (completely not true) to 5 (completely true), where higher scores indicate a more positive therapeutic relationship. Cronbach’s α for therapeutic relationship was 0.903.

#### 2.2.3. Therapeutic Satisfaction

Therapeutic satisfaction was measured using the scale of the Client Satisfaction Questionnaire (CSQ) which was reconstructed by Im [30]. The scale consists of seven items measuring the extent to which the user is satisfied with their community mental health service, such as if the service has helped them understand their problem, get along with others, meet their needs, or manage their symptoms. Each item was rated on a scale from 1 (completely not true) to 5 (completely true), where higher scores indicate higher levels of satisfaction with community mental health services. Cronbach’s α for therapeutic satisfaction was 0.915.

#### 2.2.4. Life Satisfaction

Life satisfaction has been identified as a distinct construct representing a cognitive and global evaluation of one’s quality of life as a whole. This was measured using the Life Satisfaction Scale, which was used in the Korea Welfare Panel Study [31]. The scale includes seven items measuring different types of satisfaction, such as satisfaction with health, family income, housing environment, family relationship, career, leisure life, and overall satisfaction. A 5-point Likert-style response scale (ranging from 1 = completely unsatisfactory to 5 = completely satisfactory) was used. Higher scores indicate higher levels of satisfaction with life. Cronbach’s α for life satisfaction was 0.900.

#### 2.2.5. Clinical Variable

Symptoms are evaluated based on the degree to which patients have recently experienced psychiatric symptoms. To measure symptoms, we used the Colorado Symptom Index, which was adapted to the Korean context by Lee and Seo [32]. This scale consists of 14 items that evaluate the extents to which patients experience hallucinations, delusions, memory loss, suicidal thinking, and mood disorders. Each question is measured on a 5-point scale ranging from “not at all (1 point)” to “strongly agree (5 points)”, and the higher the score, the higher the level of symptoms. Cronbach’s α for symptoms was 0.914.

Insight is the ability of patients to accept that they have a mental illness and to recognize both the symptoms of that mental illness and the need for treatment. Insight was assessed using the SAI (Schedule for the Assessment of Insight) [33], which was translated to Korean by Byeon [34]. This scale consists of five items, such as awareness of one’s illness, the capacity to recognize relapse psychotic symptoms as abnormal, and treatment compliance. Each question is answered on a 5-point scale ranging from “not at all (1 point)” to “strongly agree (5 points)”, and higher scores indicate higher levels of insight. Cronbach’s α for insight was 0.766.

Psychosocial functioning refers to measures of physical, psychological, and social aspects that are necessary to maintain and manage a healthy life of persons with mental illness. This was measured with the Self-Health Management Scale developed by Seo [35] to assess the social functions of patients. This scale consists of 10 questions including compliance with treatment and drug use, managing stress, living regularly and socializing, and asking for appropriate professional help. Each question is answered on a 5-point scale ranging from “not at all (1 point)” to “strongly agree (5 points)”, and the higher the score, the higher the psychosocial functioning of the people with mental illness. Cronbach’s α for self-health management was 0.792.

### 2.3. Statistical Analyses

Statistical analyses were performed using SPSS 26 and AMOS 26 (SPSS Inc., Chicago, IL, USA). In order to review the basic assumptions of regression analysis before analyzing, we examined outliers, normality, and multi-collinearity. To verify the reliability of the scale, Cronbach’s α internal consistency reliability was used. Descriptive statistical analysis was conducted to examine the socio-demographic characteristics of the participants. Bivariate correlation analysis was used to analyze the correlations among the variables.

Structural equation modeling (SEM), which involves a measurement model and a structural model, was used to analyze the cause and effect relations between latent variables. Observed variables are used to measure latent variables in measurement models, and the relationships between latent variables are tested in the structural model [36]. Using a two-step approach, confirmatory factor analysis (CFA) was first conducted to test the validity of the measurement models. The next step was to test and improve the goodness-of-fit of the structural model using modification indices. The absolute fit measure and incremental fit measures were considered together. After the optimized model was derived and then confirmed, the significant influencing factors and the regression weights were evaluated.

## 3. Results

### 3.1. Bivariate Association between Clinical Variable and Perceived Coercion

Table 2 presents the results of a correlation analysis conducted to examine the relationship between the clinical and main variables. Symptoms were found to have negative correlations with the major variables, while insight and psychosocial functioning had positive correlations. That is, the more symptoms there were, the more the therapeutic relationship (r = −0.178), therapeutic satisfaction (r = −0.229), and life satisfaction (r = −0.566) tended to decrease. Conversely, the higher the psychosocial functioning, the more the therapeutic relationship (r = 0.491), therapeutic satisfaction (r = 0.543), and life satisfaction (r = 0.495) tended to increase. The higher the insight, the higher the therapeutic relationship (r = 0.371) and therapeutic satisfaction (r = 0.308), but no significant correlation was found with life satisfaction. Additionally, there were no significant correlations between clinical variables and perceived coercion.

Perceived coercion, on the other hand, showed significant negative correlations with therapeutic relationship and therapeutic satisfaction. Namely, the higher the perceived coercion, the lower the therapeutic relationship (r = −0.233), and the lower the therapeutic satisfaction (r = −0.282). However, with life satisfaction, no significant correlation was found.

### 3.2. Confirmatory Factor Analysis

Confirmatory factor analysis (CFA) was conducted to assess the observed variables with factor loadings less than 0.5 and to confirm the fitness of the measurement model for each of the observed variables of perceived coercion, therapeutic relationship, therapeutic satisfaction, and life satisfaction. All factor loading of the observed variables was over 0.6, as shown in Figure 1. The CFA for all constructions was used without removing the observed variables. It can be seen that there was a total of 16 observed variables in the measurement model. The goodness-of-fit indices of the full measurement model are presented in Table 3. From Table 3, the chi-square test statistic was found to be inappropriate, but other indices might be considered together. Most of the indices met the corresponding acceptable requirements.

### 3.3. Research Model Verification

The initial research model analysis results did not fit the data very well. In order to modify the initial research model, correlations were made between the within-factor measurement error of the observed variables as the modification indices suggested. The final research model was derived as shown in Figure 2. The goodness-of-fit of the final research model was evaluated to verify the effect of perceived coercion on therapeutic satisfaction and life satisfaction mediated by therapeutic relationships, as shown in Table 4. From Table 4, the chi-square test statistic was found to be inappropriate, but other indices might be considered together. Most of the indices of the final model showed sufficient goodness-of-fit.

From Table 5, it can be seen that three paths from PC to TR, from TR to TS, and from TR to LS are significant at the 0.05, 0.001, and 0.001 levels, respectively. Therefore, perceived coercion significantly affected therapeutic satisfaction and life satisfaction mediated by the therapeutic relationship. The path regression weight from PC to TR was −0.216, from TR to TS it was 0.762, and from TR to LS it was 0.271. In other words, the lower perceived coercion is, the better the therapeutic relationship is, and this has a positive effect on the therapeutic satisfaction and life satisfaction of persons with mental illness.

The result of examining the relative influence of each variable through standardized regression weights was that the therapeutic relationship showed the greatest impact on therapeutic satisfaction. This was followed by the effect of the therapeutic relationship on life satisfaction, and the effect of the perceived coercion on the therapeutic relationship.

## 4. Discussion

The purpose of this study was to examine how perceived coercion in the community treatment process (e.g., coercion in hospitalization) influences therapeutic satisfaction and life satisfaction mediated by the therapeutic relationship. Two important findings were identified.

First, the perceived coercion in the community does not have a meaningful correlation with the subject’s symptoms, psychosocial functioning, or insight. These results differ from those indicating that the perceived coercion experienced during hospitalization is higher since symptoms are more severe, psychosocial functioning is lower, and insight is lower [7,23,37]. However, these results are potentially controversial, as some studies [5,38] have shown that symptoms and psychosocial functioning are not related to perceived coercion, even in the hospitalization process. The finding that perceived coercion is not associated with disease-related characteristics refutes the premise of a paternalistic approach. The paternalistic perspective assumes that coercion against the intention of the parties is justified for the therapeutic benefit of patients with severe symptoms and a lack of insight. However, the results of this study indicate that the paternalistic premise is not supported, as perceived coercion does not have a meaningful relationship with symptoms or insight.

Second, perceived coercion was shown to negatively affect the therapeutic relationship, and this in turn has significant effects on the patient’s therapeutic satisfaction and life satisfaction. This is consistent with the research results indicating that coercion negatively affects the therapeutic relationship [23,39,40,41], thereby negatively affecting therapeutic satisfaction and life satisfaction [14,42,43,44,45]. Similar to the consequences of perceived coercion in hospitalization, these results show that the coercion experienced in the community negatively affects the relationship with the therapist and hinders the formation of a trusting relationship with them. The voluntary participation of the parties is vital for continued treatment and recovery in the community that presupposes the least restrictiveness. However, a negative relationship with the therapist may hinder continuous treatment by lowering an individual’s voluntary will towards therapy, and in turn reducing therapeutic satisfaction. Moreover, having a negative relationship with one’s therapist, who is an important source of psychosocial support in a patient’s life, is potentially hazardous in that it can delay recovery by decreasing life satisfaction.

## 5. Conclusions

Based on these results, the authors propose strategies to minimize coercion in community-based treatment. There is no “perfect model in which coercion is absent” [46]; however, there are strategies that can be used to fully minimize the use of coercion. One of the main strategies is to use “psychiatric advance directives”. This allows individuals to clearly state their preference for future treatment at times when they may not be able to make considered decisions [47]. This can give an individual control over their treatment in the long run, thus minimizing coercion. In addition, an empowerment strategy that provides a variety of information and resources and that helps an individual freely express their needs and emotions (e.g., righteous anger) can not only minimize coercion, but also help form a positive therapeutic relationship [46]. However, rapid clinical and risk assessments are the most important things. In contrast to the situation in hospitals, wherein continuous observation is possible, accurate and rapid assessment is a means of ensuring maximum autonomy and preventing more serious coercion such as involuntary hospitalization.

This study found that coercion perceived by the persons with mental illness in the community, as well as the coercion in the hospital, could seriously damage their quality of life. These results may contribute to increasing the interest of researchers and practitioners in community coercion, which was of relatively less interest than coercion in hospitals. However, as a limitation of this study, we identified first that longitudinal study is needed to analyze the impact of perceived coercion on quality of life. Insight develops over time; so at first, from a paternalistic point of view, the individual may perceive coercion as unpleasant but, as symptoms improve, they may eventually come to appreciate coercion. Thus, even if negative results appear in the short term, they may change over time. Therefore, the impact of coercion can be clearly confirmed when we track changes. Second, this study focused only on perceived coercion, which is a subjective coercion, and failed to analyze the relationship with objective coercion. Since perceived coercion is based on individual perception, the individual has the advantage of being able to evaluate its influence on themselves in the most meaningful way possible, but there are limitations that can be exaggerated or distorted. It is therefore necessary to consider subjective coercion and objective coercion together.

## Figures and Tables

**Figure 1 ijerph-18-02290-f001:**
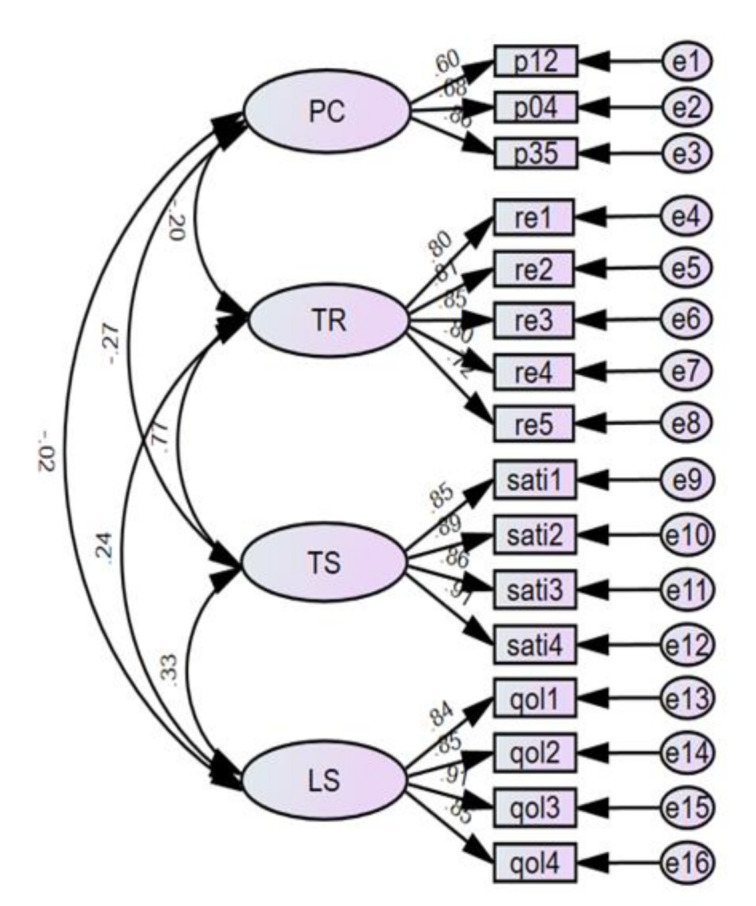
Standardized regression weights of the full measurement model. PC: perceived coercion, TR: therapeutic relationship, TS: therapeutic satisfaction, LS: life satisfaction.

**Figure 2 ijerph-18-02290-f002:**
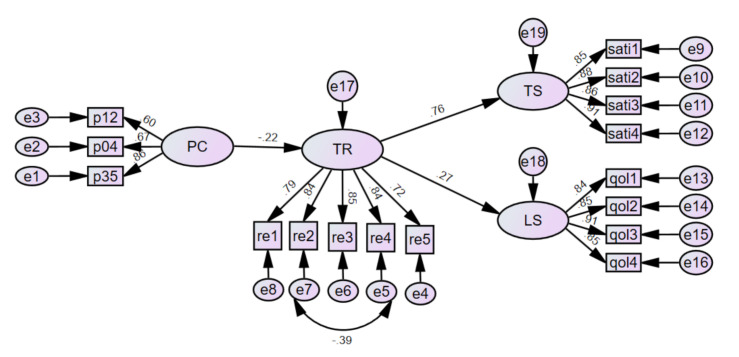
Standardized regression weights of the final research model. PC: perceived coercion, TR: therapeutic relationship, TS: therapeutic satisfaction, LS: life satisfaction.

**Table 1 ijerph-18-02290-t001:** Socio-demographic characteristics of the participants.

Variable	Category	Frequency	Percentage (%)
Gender	Male	98	53
	Female	85	45.9
	No response	2	1.1
Age	20–29	18	9.7
	30–39	31	16.8
	40–49	46	24.9
	50–59	56	30.3
	Over 60	34	18.4
Educational level	Middle school or below	41	22.2
	High school	99	53.5
	University or over	41	22.2
	Others	2	1.1
Diagnosis	Schizophrenia	119	64.3
	Major depression	44	23.8
	Bipolar disorder	13	7.0
Facility type	Mental health center	93	50.3
	Rehabilitation center	92	49.7
Employment status	Vocational rehabilitation	5	10.6
	Part time	17	36.2
	Full time	15	31.9
	Others	10	21.3
Housemate	Alone	62	33.5
	Family	109	58.9
	Peer	9	4.9
	Others	2	1.1

**Table 2 ijerph-18-02290-t002:** Correlations among variables and perceived coercion.

	1	2	3	4	5	6
1. Symptom	-					
2. Insight	−0.057	-				
3. Psychosocial functioning	−0.485 **	0.349 **	-			
4. Perceived coercion	−0.127	−0.112	−0.133	-		
5. Therapeutic relationship	−0.178 *	0.371 **	0.491 **	−0.233 **	-	
6. Therapeutic satisfaction	−0.229 **	0.308 **	0.543 **	−0.282 **	0.702 **	-
7. Life satisfaction	−0.566 **	0.061	0.495 **	−0.030	0.218 **	0.302 **

**p* < 0.05, ** *p* < 0.01. 1, 2, 3: clinical variables.

**Table 3 ijerph-18-02290-t003:** Goodness-of-fit of the measurement model.

$x^{2}$ = 164.822 (*p* < 0.001), Degrees of Freedom = 98 (136–138)			
Goodness-of-Fit Measure		Level of Acceptance Fit	Fit Statistic
Absolute fit	$x^{2}$/df	<3 good	1.682
	GFI	>0.8 acceptable, >0.9 good	0.901
	AGFI	>0.8 acceptable, >0.9 good	0.862
	RMSEA	<0.08 good	0.061
Incremental fit	NFI	>0.9 good	0.922
	RFI	>0.9 good	0.904
	IFI	>0.9 good	0.967
	TLI	>0.9 good	0.959
	CFI	>0.9 good	0.966

GFI: goodness-of-fit-index, AGFI: adjusted goodness-of-fit-index, RMSEA: root mean square error of approximation, NFI: normed fit index, RFI: relative fit index, IFI: incremental fit index, TLI: Tucker-Lewis index, CFI: comparative fit index.

**Table 4 ijerph-18-02290-t004:** Goodness-of-fit of the final research model.

$x^{2}$ = 163.431 (*p* < 0.001), Degrees of Freedom = 100 (136–136)			
Goodness-of-Fit Measure		Level of Acceptance Fit	Fit Statistic
Absolute fit	$x^{2}$/df	<3 good	1.634
	GFI	>0.8 acceptable, >0.9 good	0.902
	AGFI	>0.8 acceptable, >0.9 good	0.867
	RMSEA	<0.08 good	0.059
Incremental fit	NFI	>0.9 good	0.923
	RFI	>0.9 good	0.907
	IFI	>0.9 good	0.968
	TLI	>0.9 good	0.962
	CFI	>0.9 good	0.968

GFI: goodness-of-fit-index, AGFI: adjusted goodness-of-fit-index, RMSEA: root mean square error of approximation, NFI: normed fit index, RFI: relative fit index, IFI: incremental fit index, TLI: Tucker-Lewis index, CFI: comparative fit index.

**Table 5 ijerph-18-02290-t005:** Regression weights in the final research model.

	B	β	S.E.	C.R.	*p*
PC→TR	−0.168	−0.216	0.068	−2.483	0.013
TR→TS	0.814	0.762	0.091	8.987	<0.001
TR→LS	0.378	0.271	0.112	3.379	<0.001

B: regression weights, β: standardized regression weights, S.E.: standard error, C.R.: critical ratio, *p*: *p*-value. PC: perceived coercion, TR: therapeutic relationship, TS: therapeutic satisfaction, LS: life satisfaction.

## Data Availability

Restrictions apply to the availability of these data. Data was obtained from the participants and are available with the permission of the participants.

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
