# Peer review of "Perceived Coercion of Persons with Mental Illness Living in a Community"

_ijerph, 2021, doi:10.3390/ijerph18052290_

Round 1

Reviewer 1 Report

This paper analyzed the effect of the perceived coercion of people with mental illness living in a community on their therapeutic satisfaction and life satisfaction mediated by therapeutic relationship. Results showed that clinical variables had statistically significant correlations with therapeutic relationship, therapeutic satisfaction, and life satisfaction, but no significant correlation with perceived coercion. Overall, I think the article expressed some interesting research findings; however, significant revisions are suggested before it can reach the publishable quality. Detailed comments are given as follows:

  1. The title of this paper should be modified to indicate the research aim of this paper more clearly.
  2. The authors should summarize the main contributions of this paper in Section 1.
  3. In Section 1, "Community treatment orders (CTOS)" is not explained clearly.
  4. Some of the abbreviations and indicators in the article were not explained, such as the meaning of SD in the Abstract and the meaning of Cronbach's indicator in Chapter 2, which should be explained to the readers to facilitate their reading.
  5. In Section 2.1, the authors are suggested to use more figures and tables to present data statistics more clearly, such as the information on the demographic characteristics of the participants.
  6. In Section 2.3, no explanation of why the method was used, and no detailed description of the experimental method.
  7. Due to the special nature of the mental state of people with mental illness, is it possible to guarantee that they were awake to participate in the experiment? Relevant information and data should be provided to prove this.
  8. More information concerning the modeling process should be provided in the methodology and results sections. It is highly recommended that the authors can refer to this paper: Investigating the determinants of contractor’s construction and demolition waste management behavior in Mainland China, which is also using SEM to investigated the relationships between constructs.
  9. Figure 1 can be more concise and intuitive.
  10. The table in the article could have further modifications. For example, Symptoms, Insight, Psychosocial functioning belong to the sub-indicators of the Clinical variable in section 2.2.5, these can be reflected in Table 1.
  11. The article should add a Conclusion section to briefly explain the results of the study and to make clear and important recommendations at the end of the article.

Author Response

Thank you for the reviewer's constructive comments. 

We revised the manuscript. 

Response to Reviewer 1 Comments

First of all, we would like to express our appreciation to the reviewer and editor who provided some constructive suggestions to improve our manuscript. We also received some critical comments, which helped us to improve our manuscript and make it suitable for publication. Hereafter, we attach our point-by-point responses to the reviewer’s comments. The red parts in the revised manuscript are the changes.

Point 1: the meaning of SD in the Abstract, the meaning of Cronbach's indicator, and explanation of why the method was used, which should be explained to the readers to facilitate their reading.

Our response: In the revised manuscript, we added the meaning and the sentence in the Abstract (p.1) and Section 2.3 (p.5).

Point 2: the authors are suggested to use more figures and tables to present data statistics more clearly

Our response:  We added Table 1 in the Section 2.1 (p.3), Figure 1 and Table 3 in the Section 3.2 (p.6), Table 4 in the Section 3.3 (p.7).

Point 3: More information concerning the modeling process should be provided in the methodology and results sections.

Our response: We added a new section 3.2 (Confirmatory factor analysis, p.6), supplemented the sentence in Section 3.3 (p.7-8).

Point 4: The table in the article could have further modifications. Figure 1 can be more concise and intuitive.

Our response: We revised the Table 2 (p.6), Figure 2 (p.7), and Table 5 (p.8).

Point 5: The authors should summarize the main contributions of this paper and the article should add a Conclusion section.

Our response: We added a new Section 5 (Conclusion, p.8-9), and summarized the contributions of this paper in that section (p.9).

Point 6: "Community treatment orders (CTOS)" is not explained clearly.

Our response: We added explanation of CTO (p.2).

Point 7: Is it possible to guarantee that they were awake to participate in the experiment?

Our response: We recruited the participants living within the community. They were maintaining ongoing contact with the community service unit. We reconfirmed with practitioners whether they were able to properly understand and consent to this study, as IRB suggested.

Reviewer 2 Report

Dear authors, first of all congratulations for this amazing study. 

Your manuscript is very well organized and it has a high significance in the content. 

Some litle suggestions that can improve your manuscript:

  1. In the results point put a litle introduction explaining how you will organize this point to present the different domains of results. 
  2. Present the description of the results after the tables. You can introduce the table. Then present the table and next presenting the description. 
  3. In the discussion, maybe it would be better not say "As with any study, there are several limitations", just say, "As limitations of this study we identify..." - It is a value judge in a generalization to all studies, althought you are right in most of the studies. 
  4. Give more enphasis in the discussion to the contributions of your study to research, to society and to clinical practices.

These are few suggestions. A litle more effort and your manuscript will be perfect. 

Author Response

Thank you for the reviewer's constructive comments.

We revised the manuscript. 

Response to Reviewer 2 Comments

First of all, we would like to express our appreciation to the reviewer and editor who provided some constructive suggestions to improve our manuscript. We also received some critical comments, which helped us to improve our manuscript and make it suitable for publication. Hereafter, we attach our point-by-point responses to the reviewer’s comments. The red parts in the revised manuscript are the changes.

Point 1: In the discussion, just say, "As limitations of this study we identify..." - It is a value judge in a generalization to all studies, although you are right in most of the studies.

Our response: The second paragraph of conclusion, we changed the sentence as suggested by the reviewer (p.9).

:  However, as limitation of this study we identify ~

Point 2: put domain of results and give more emphasis in the discussion to the contributions of your study

Our response: We added a new Section 5 (Conclusion, p.8-9), and summarized the contributions of this paper in that section (p.9).

Reviewer 3 Report

Thank you for the opportunity to review ‘Perceived Coercion of Persons With Mental Illness Living in a Community’. This is a well-structured and mostly well-written paper that considers the relationship between perceived coercion of people with mental illness living in the community and their therapeutic and life satisfaction. I think this is an interesting paper. I have few concerns with the methodology or data analysis per se. I do have some concerns about the research question and the framework of analysis. I think there is an ethical dilemma that underpins this paper, but it is not the one that the authors identify in the Introduction (coercive intervention, therapeutic alliance, and therapeutic gains). The ethical dilemma for me is whether or not the authors are in fact truly independent researchers who are in a position to critique the way mental health care is provided in Korea, given that the funder of the research appears to be the Korean Ministry of Education and the NRF of Korea, both, presumably, government sources.

The authors use the notion of ‘perceived coercion’ in a way that implies that the coercion is only perceived, not actual. If persons with mental health challenges perceive that they are being coerced, it is very likely that they are being coerced. Coercion should only be used when there is a genuine risk to self or others; the enforced stay should be no longer than 72 hours, and it should be reviewed by a judge if there is a clinical argument to extend that stay; that is the standard in most countries. The authors attribute coercion in Korean mental health treatment to ‘paternalism’, but do not define what paternalism means in this context (although they suggest ‘authoritarian’). If paternalism means that government, or the received model of mental health care, is coercive, then perception is reality. The authors attribute the acceptance of ‘authoritarian or paternalistic approaches more easily’ than the West because of the collectivist culture in Korea, but there are no comparisons with similar cultural contexts such as Japan (which halved its mental health inpatient census from 2014, although it has the highest number of psychiatric beds of any OECD nation; Kanata, 2016) or PRC.

Korea has the highest rate of suicidality in the OECD, the highest rate of alcohol consumption in the world, and one of the highest mental health inpatient hospitalisation rates in the world (Watkins, 2018). Surely these are symptoms of much broader and deeper social issues that include how mental health is conceptualised by the public (and government), and recognised, diagnosed, and treated by professionals. (Compulsory inpatient treatment for mental health issues conjures up memories of how mental health ‘diagnoses’ and compulsory ‘treatment’ were used to control dissidents in the USSR and PRC a few decades ago.) Successful suicides in Korea had frequently sought prior treatment for somatic rather than mental health symptoms, and mental health issues such as depression were not diagnosed or treated. This reviewer is prompted to ask whether the Korean government sets compulsory (coercive) inpatient care as a way to manage the reputational damage suicidality, alcohol use, stress and depression create for Korea.

There is no doubt that mental health disorders (and the people who live with these disorders) are stigmatised both in Korea and in the Korean diaspora—I have supervised students who have researched this stigma, and the literature is also clear on this issue, as is the present paper. But regardless of how elegant the statistical analyses or model are in the present paper, to publish a paper that merely recommends minimising ‘perceived coercion’ (p. 7) raises serious ethical concerns for me. But I also wonder whether the Korea government is prepared to be challenged on its approach. That, then, is the real ethical dilemma that underpins this paper. Are the authors prepared to ignore that dilemma? Is IJERPH prepared to ignore that dilemma? I hope not.

I hope that the authors will not take this challenge as disapproving of their work: they have done what appears to be good work with solid statistical analysis and the paper is, as noted, well-presented. I think there is a way to reframe their findings that invites and encourages alternative pathways for mental health management in Korea, including public information and awareness campaigns, early intervention programmes in schools and workplaces, working with physicians, nurses and other front-line care providers to recognise, diagnose, and provide appropriate  treatment or referrals for people with mental health disorders, and similar. (Even my students in an Anglophone nation know that many Asians will present with somatic symptoms rather than mental health disorders.) All of this will take resourcing. Minimising the perception of coercion does not make the coercion go away. The authors acknowledge that a positive therapeutic relationship with a care provider is essential for a good outcome for people with mental health disorders. The authors have an opportunity to help the Korean government to develop an alternative, truly non-coercive but ultimately much more effective long-term mental health strategy, one that will have good outcomes for individuals, families, and the community (see the work of Kyooseob HA at Seoul National University). I encourage the authors to revise the introduction as well as the conclusion and recommendations section of this paper to identify more clearly the ethical challenges raised by their work, and possible pathways forward.

Specific comments

‘Paternalism’ used throughout must be clearly defined in the context of the topic and this paper.

on p. 6, first paragraph of the Discussion section, ‘facts’ is not the right word. I would use the word ‘findings’.  

Author Response

Thank you for the reviewer's constructive comments.

We revised the manuscript. 

Response to Reviewer 3 Comments

First of all, we would like to express our appreciation to the reviewer and editor who provided some constructive suggestions to improve our manuscript. We also received some critical comments, which helped us to improve our manuscript and make it suitable for publication. Hereafter, we attach our point-by-point responses to the reviewer’s comments. The red parts in the revised manuscript are the changes.

Point 1: explain the notion of ‘perceived coercion’ in this article

Our response: The second paragraph of introduction, we added explanation of perceived coercion (p.1).

Point 2: ‘Paternalism’ used throughout must be clearly defined in the context of the topic and this paper.

Our response: We added definition of paternalism (p.2).

Point 3: that merely recommends minimizing ‘perceived coercion’ (p.7) raises serious ethical concerns for me.

Our response: We revised the words in the Abstract (p.1) and Section 5 (p.8-9).

: to minimize coercion ~

Point 4: on p. 6, first paragraph of the Discussion section, ‘facts’ is not the right word. I would use the word ‘finding’.

Our response: We changed ‘facts’ to ‘finding’ in Section 4 (p.8) as suggested by the reviewer

Round 2

Reviewer 1 Report

Most of my comments have been addressed. However, I suggest minor revisions should be made before this paper can be published. The current abstract is not following the requirements of IJERPH. The abstract should be a single paragraph and should follow the style of structured abstracts, but without headings. The authors can refer to https://www.mdpi.com/journal/ijerph/instructions for details. 

Reviewer 3 Report

I acknowledge the responsiveness of the authors to reviewer feedback. The presentation of the statistical analysis is clearer and more transparent. The revisions to the abstract, introductions and conclusion are helpful and responsive to reviewer feedback. I would like to have seen the authors take the next step to challenge what appears to be a coercive mental health system in Korea, but perhaps this is not possible, wise or culturally appropriate. Nevertheless at some point such a challenge will be important if Korean mental health care is to come into the 21st century. I commend the authors on their important work.